# Deriving the Dietary Approaches to Stop Hypertension (DASH) Score in Women from Seven Pregnancy Cohorts from the European ALPHABET Consortium

**DOI:** 10.3390/nu11112706

**Published:** 2019-11-08

**Authors:** Adrien M. Aubert, Anne Forhan, Blandine de Lauzon-Guillain, Ling-Wei Chen, Kinga Polanska, Wojciech Hanke, Agnieszka Jankowska, Sara M. Mensink-Bout, Liesbeth Duijts, Matthew Suderman, Caroline L. Relton, Sarah R. Crozier, Nicholas C. Harvey, Cyrus Cooper, Fionnuala M. McAuliffe, Cecily C. Kelleher, Catherine M. Phillips, Barbara Heude, Jonathan Y. Bernard

**Affiliations:** 1Centre for Research in Epidemiology and StatisticS (CRESS), Université de Paris, Inserm, Inra, F-75004 Paris, France; adrien.aubert@inserm.fr (A.M.A.); anne.forhan@inserm.fr (A.F.); blandine.delauzon@inserm.fr (B.d.L.-G.); jonathan.bernard@inserm.fr (J.Y.B.); 2HRB Centre for Health and Diet Research, School of Public Health, Physiotherapy, and Sports Science, University College Dublin, Belfield, Dublin 4, Ireland; ling-wei.chen@ucd.ie (L.-W.C.); cecily.kelleher@ucd.ie (C.C.K.); catherine.phillips@ucd.ie (C.M.P.); 3Department of Environmental Epidemiology, Nofer Institute of Occupational Medicine, 91-348 Lodz, Poland; kinga.polanska@imp.lodz.pl (K.P.); wojciech.hanke@imp.lodz.pl (W.H.); agnieszka.jankowska@imp.lodz.pl (A.J.); 4The Generation R Study Group, Erasmus MC, University Medical Center Rotterdam, 3000 CA Rotterdam, The Netherlands; s.mensink-bout@erasmusmc.nl (S.M.M.-B.); l.duijts@erasmusmc.nl (L.D.); 5Department of Pediatrics, Division of Respiratory Medicine and Allergology, Erasmus MC, University Medical Center Rotterdam, 3000 CB Rotterdam, The Netherlands; 6Department of Pediatrics, division of Neonatology, Erasmus MC, University Medical Center Rotterdam, 3000 CB Rotterdam, The Netherlands; 7MRC Integrative Epidemiology Unit, Population Health Sciences, Bristol Medical School, University of Bristol, Bristol BS8 2BN, UK; matthew.suderman@bristol.ac.uk (M.S.); caroline.relton@bristol.ac.uk (C.L.R.); 8MRC Lifecourse Epidemiology Unit, University of Southampton, Southampton General Hospital Southampton, Southampton SO16 6YD, UK; src@mrc.soton.ac.uk (S.R.C.); nch@mrc.soton.ac.uk (N.C.H.); cc@mrc.soton.ac.uk (C.C.); 9NIHR Southampton Biomedical Research Centre, University of Southampton and University Hospital Southampton NHS Foundation Trust, Southampton SO16 6YD, UK; 10NIHR Oxford Biomedical Research Centre, University of Oxford, Oxford OX1 2JD, UK; 11UCD Perinatal Research Centre, School of Medicine, University College Dublin, National Maternity Hospital, Dublin 2, Ireland; fionnuala.mcauliffe@ucd.ie; 12Singapore Institute for Clinical Sciences (SICS), Agency for Science, Technology and Research (A*STAR), 117609 Singapore, Singapore

**Keywords:** DASH (Dietary Approaches to Stop Hypertension), diet, nutrition, pregnancy

## Abstract

The ALPHABET consortium aims to examine the interplays between maternal diet quality, epigenetics and offspring health in seven pregnancy/birth cohorts from five European countries. We aimed to use the Dietary Approaches to Stop Hypertension (DASH) score to assess diet quality, but different versions have been published. To derive a single DASH score allowing cross-country, cross-cohort and cross-period comparison and limiting data heterogeneity within the ALPHABET consortium, we harmonised food frequency questionnaire (FFQ) data collected before and during pregnancy in ≥26,500 women. Although FFQs differed strongly in length and content, we derived a consortium DASH score composed of eight food components by combining the prescriptive original DASH and the DASH described by Fung et al. Statistical issues tied to the nature of the FFQs led us to re-classify two food groups (grains and dairy products). Most DASH food components exhibited pronounced between-cohort variability, including non-full-fat dairy products (median intake ranging from 0.1 to 2.2 servings/day), sugar-sweetened beverages/sweets/added sugars (0.3–1.7 servings/day), fruits (1.1–3.1 servings/day), and vegetables (1.5–3.6 servings/day). We successfully developed a harmonized DASH score adapted to all cohorts being part of the ALPHABET consortium. This methodological work may benefit other research teams in adapting the DASH to their study’s specificities.

## 1. Introduction

The Developmental Origins of Health and Disease (DOHaD) paradigm suggests that environmental exposures during critical periods of early life development, even before conception, may influence later health in childhood and adulthood [1,2]. It has been established that maternal diet is an important early-life exposure and determinant of both maternal, neonatal and child health outcomes [3]. Indeed, interventional and observational studies have highlighted the role of certain macronutrients, micronutrients, and vitamins on pregnancy complications (e.g., decreased risk of pre-eclampsia), birth outcomes (e.g., decreased risk of preterm birth and low birth weight), and offspring health and growth (e.g., obesity, respiratory health and neurocognitive development) [4,5,6,7,8,9]. Despite extensive research on the potential importance of maternal intakes of individual nutrients and food groups, looking at dietary quality holistically based on whole diet can improve applicability for public health messaging, since people do not consume nutrients in isolation.

Several scales or indices have been proposed to measure diet quality, and among them the Dietary Approaches to Stop Hypertension (DASH) score is commonly used. Indeed, numerous studies have related the adherence to a DASH diet to several health outcomes such as cardiovascular and metabolic disorders [10,11,12], cancers [13] or weight management [14]. For instance, a meta-analysis on cardiovascular risk factors concluded that the DASH diet was *“an effective nutritional strategy to prevent cardiovascular diseases”* [15]. Specifically, among pregnant women with complications such as gestational or chronic hypertension and gestational diabetes mellitus, favouring a DASH diet can be a potential strategy for improving pregnancy outcomes [16,17]. However, among women without pregnancy complications, the protective effects of the DASH diet against risk of adverse pregnancy and birth outcomes are inconsistent, warranting further research in general pregnant populations [18].

Gathering data from seven mother-child cohorts from five European countries, the ALPHABET consortium aims to expand the knowledge base regarding the interplay between maternal diet quality (defined by the DASH score), dietary inflammation (defined by the dietary inflammatory index [19]), epigenetics (DNA methylation), and offspring health (adiposity, bone, cardiometabolic, respiratory and neurodevelopmental health) and identify biomarkers that may inform future public health strategies. To examine these research questions, the DASH score, used to assess maternal dietary quality, needs to be derived in a harmonised way for the seven ALPHABET cohorts to reduce heterogeneity. The harmonisation is important because even though it is common to assess dietary intake through food frequency questionnaires (FFQ) in epidemiological studies, these questionnaires can differ in structure and length from one cohort to the other. Some studies have successfully achieved post hoc standardisation of dietary data from diverse sources and cohorts [20,21]. Furthermore, several variants of DASH index derivation methods have been published, with notable differences in the food components included and the scoring criteria [15,16]. No consensus exists but the most commonly used DASH score method is the one proposed by Fung et al. [10], which ranks participants based on quintiles of dietary intakes.

In this context, the present work aims to (1) explain the method used to derive a harmonised DASH score in the ALPHABET consortium and detail the challenges encountered during this process and (2) describe the maternal DASH score and related food consumption in different cohorts and over three periods of assessment: pre-pregnancy, early pregnancy, and late pregnancy.

## 2. Materials and Methods 

### 2.1. Study Populations

The ALPHABET project is a European consortium comprised of seven longitudinal birth cohort studies: the Avon Longitudinal Study of Parents and Children (ALSPAC), the study on the pre- and early postnatal determinants of child health and development (EDEN), the Generation R Study (Generation R), the Lifeways Cross-Generation Cohort Study (Lifeways), the Polish Mother and Child Cohort (REPRO_PL), the Randomised Control Trial of Low Glycaemic Index Diet study (ROLO), and the Southampton Women’s Survey (SWS). Details on study designs and sample sizes of these cohorts are available elsewhere [22,23,24,25,26,27,28,29,30].

Descriptive characteristics of each of the cohorts within the ALPHABET consortium are presented in Table 1. Of the seven included studies, two were based in Ireland, two in England, and one each in Poland, the Netherlands, and France. Most women were recruited during pregnancy, except for SWS where recruitment commenced before pregnancy. Started in 1990, ALSPAC was the oldest study, whereas REPRO_PL and ROLO, both started in 2007, were the most recent. The sample size ranged from 759 (ROLO) to 14,541 (ALSPAC). Since all studies assessed maternal diet with FFQs at times which differed between cohorts, we categorised into three periods: pre-pregnancy, early pregnancy (1st or 2nd trimester) and late pregnancy (3rd trimester).

All participating cohorts have obtained the relevant institutional ethical approval and research to date has been conducted according to the guidelines laid down in the Declaration of Helsinki.

### 2.2. Dietary Data Collection and Treatment

Women completed mostly validated (except ALSPAC), semi-quantitative (EDEN [31], Generation R [32], Lifeways [33,34], ROLO [34,35]), or non-quantitative (ALSPAC [36], REPRO_PL [28], SWS [37]) FFQs, which were designed to assess average dietary intake over pre-conception or pregnancy periods (Table 1). Women declared food intake on frequency scales ranging from five (in ALSPAC) to nine response categories (in Generation R, Lifeways and ROLO) (Appendix A). An item “not ticked” (missing) was considered as “non-consuming” and imputed with zero, assuming that these mothers did not eat it. All food consumption frequencies were converted into daily frequencies (servings per day) to be comparable across cohorts. For cohorts with semi-quantitative FFQs, we also calculated food consumption data in amounts (grams and millilitres per day).

### 2.3. DASH Score Creation

Several DASH scores have been developed or adapted in the literature, which differ regarding both the food components included and scoring method [38]. The DASH diet was initially created to help reduce arterial hypertension but no consensus exists on how to generate DASH scores from FFQs [39]. To our knowledge, the DASH index proposed by Fung et al. [10] has been the most widely used. Therefore, we generated DASH scores, using the Fung method, from the data collected within each of the ALPHABET consortium cohorts and adapted to their specificities as described below. Previously published DASH scores were based on whether one meets a recommended minimum number of servings [40]. In contrast, Fung’s DASH index relies on quintile ranking, allowing for a wider, more-discriminating score range [38], an approach that we judged more appropriate for ALPHABET considering the diversity of cohorts, time periods, and FFQs used. Because the main purpose of a FFQ is to rank participants according to their reported intakes rather than estimation of absolute intakes (sodium in particular is not estimated well with FFQ), scoring by quintiles would be less prone to misclassification [10].

#### 2.3.1. Food Group Classification and Item Selections

To select food groups and to classify food items into food components, we referred to the Fung’s DASH [10], the original “DASH Eating Plan” [41] and the Eurocode 2 [42]. Table 2 presents an inventory of the number of food items available by food component between cohorts, based on food components included in the Fung’s DASH or the original “DASH Eating Plan”. As illustrated, the details for dietary data differed across the ALPHABET consortium cohorts. The number of FFQ items ranged from 43 to almost 300. The ALSPAC FFQ included less than 8 food items within each food component. In contrast, the Generation R questionnaire included more than 8 food items for most food groups (except for sugar-sweetened beverages). The number of items varied also according to food components: vegetables (mean = 18.6), fruits (mean = 11.9), and red and processed meats (mean = 12.0) were generally assessed by more than 11 food items, while whole grains (mean = 5.3) and low-fat dairy products (mean = 4.3) food component were assessed through a much smaller number of items. A food component with a limited number of items included may result in statistical distributions that are less able to discriminate participants from each other. Therefore, we inventoried all relevant food items in all cohorts and examined all food component distributions.

#### 2.3.2. Scoring Method

For each food component, consumption frequency was divided into quintiles within each cohort, and participants were then classified according to their intake ranking. Consumption of food components with a high recommended intake was rated on a scale from 1 to 5 using their quintile number such that participants in Quintile 1 (lowest consumption) received a score of 1, and those in Quintile 5 (highest consumption) received a score of 5. Conversely, dietary components with a low recommended intake were scored on a reverse scale with lower consumption receiving a higher score. Finally, component scores were summed up and an overall DASH score for each participant was calculated. A higher score characterizes a higher dietary quality.

### 2.4. Statistical Analyses

The different DASH food component consumption (in frequencies and/or amounts) were described for each cohort and time period (pre-pregnancy, early pregnancy and late pregnancy) using the median (interquartile range, IQR) and compared graphically using the quintiles of the distributions. Spearman’s correlations were calculated for both DASH scores based on frequencies with those based on amounts (for cohorts with data available for both units). All analyses were carried out with SAS software v9.4 (SAS Institute Inc, Cary, North Carolina, USA) and RStudio was used to generate the figures.

## 3. Results

### 3.1. DASH Creation Choices—Food Components and Items Retained

Two food components, i.e., whole grain and low-fat dairy products consist of a low number of food items, a low number of response categories and a high number of non-consumers. For illustration purposes, Figure 1A displays the distributions of whole grains and total grains in the EDEN cohort, and Figure 1B displays the distributions of low-fat and non-full-fat dairy product components in the SWS cohort. Quintiles cannot be derived from such distributions and would not permit generation of a continuous score. Consequently, we reconsidered the whole grains and low-fat dairy products food component from the Fung’s DASH score into total grains and non-full-fat dairy products. Finally, the DASH score we developed within the seven cohorts of the ALPHABET consortium was composed of eight food components (seven food groups and one nutrient) (Table 3). For more details, Appendix A summarizes the included and excluded foods for each food component composing the DASH score and the corresponding score criteria. A high score corresponds to high intakes of total grains, vegetables (excluding potatoes and condiments), fruits, non-full-fat dairy products, and nuts/seeds/legumes, and low intakes of red and processed meats, sugar-sweetened beverages/sweets/added sugars, and sodium. Adapted to ALPHABET’s specificities, this DASH score was a composite score ranking from 8 to 40 points. As displayed in Table 2, we used 48.1% (Generation R) to 79.1% (ALSPAC) of the total FFQ food items (excluding alcohol) for creating the DASH score (ALPHABET consortium mean = 57.8%).

### 3.2. DASH Scores and Intakes of DASH Food Groups

By construct, mean (SD) DASH scores centred around 24 (median value between 8 and 40): from 23.7 (4.6) points (Lifeways’ score at early pregnancy from frequencies) to 24.1 (4.3) points (SWS’ score at early pregnancy from frequencies). Spearman’s correlation coefficients (rho (95% CI)) between DASH scores from frequencies and amounts were consistently very high: from 0.88 (0.86, 0.89) for ROLO to 0.92 (0.91, 0.93) for Lifeways, both at early pregnancy (Appendix A).

Table 4 presents median intakes of selected food components of the DASH score cohort and period. Consumption of nuts/seeds/legumes was relatively similar between cohorts: median values were between 0.1 and 0.3 servings/day. Other food component consumption displayed more between-cohort variability: total grains ranged from 1.6 to 3.5 servings/day, vegetables (excluding potatoes and condiments) from 1.5 to 3.6 servings/day, fruits from 1.1 to 3.1 servings/day, non-full-fat dairy products from 0.1 to 2.2 servings/day (or 18 to 417 grams/day in cohorts with data available in amounts), red and processed meats from 0.4 to 1.0 servings/day, sugar-sweetened beverages/sweets/added sugars from 0.3 to 1.7 servings/day, and lastly sodium ranged from 2.2 to 3.3 grams/day.

The daily food component intake frequencies corresponding to the quintiles of the distributions for every DASH component (except sodium consumption which are presented as grams/day) are displayed in Figure 2. Some differences in quintile distributions were observed between cohorts. For instance, for vegetables (excluding potatoes and condiments) intake, frequency consumption of quintiles 2–4 (20th to 80th percentile) ranged from 0.7 to 2.7 servings/day in EDEN and 0.9 to 2.0 servings/day in ALSPAC, which are noticeably lower than those observed in other cohorts e.g., 2.2–5.5 servings/day in ROLO. Another example is non-full-fat dairy products, for which frequency consumption of quintiles 2–4 (20th to 80th percentile) ranged from 0.1 to 1.1, 1.0 to 1.4, and 1.2 to 3.4 servings/day in Lifeways, ALSPAC and EDEN (during late pregnancy), respectively. Conversely, other food components (e.g., fruits, red and processed meats, sodium) varied less among cohorts.

## 4. Discussion

We derived DASH scores for pregnant women from seven European birth cohorts after harmonising data from FFQs of various lengths and degrees of detail. We encountered scientific and methodological issues that led us to adapt previously published DASH scores to specificities of the cohorts included in the ALPHABET consortium. The DASH score that we developed relied on eight food components and ranged from 8 to 40 points. A higher score characterizes a higher dietary quality.

### 4.1. Harmonisation Process Choices

Several studies have previously successfully harmonised dietary data from diverse sources and cohorts [20,21,47,48,49,50], demonstrating that retrospective harmonisation is possible. To the best of our knowledge, our study was the first to derive a DASH score from FFQs of different lengths and details across multiple distinct cohorts. Harmonising data was, however, not without challenges.

First, several versions of the DASH score have been used in the literature with large differences in food components included and in scoring criteria [38,39]. Indeed, some published indices were nutrient-based [51] while others had a food group-based approach that considered eight [10], nine [52], ten [53] or eleven [54] distinct food groups. Furthermore differences exist in the way scores were calculated, by comparing food group consumption to dietary guidelines (minimum or maximum intakes) [40] vs. within-study population distribution ranked into quintiles [10]. These methodological discrepancies might be explained by the fact that the DASH score was originally developed in the U.S. as a diet plan for intervention studies or preventive strategies [41,55,56] and was not specifically designed to describe dietary habits in other countries nor for epidemiological observational studies where diet is assessed using FFQs.

Second, the number of available food items and frequency scale categories of some food components resulted in a lack of data variability in some of our cohorts, complicating the derivation of balanced quintiles for whole grains and low-fat dairy products. To overcome this, we redefined whole grains as total grains and low-fat dairy products as non-full-fat dairy products. Although this resulted in slight deviations from the original DASH guidelines, we argue that it is supported by a sound rationale (Table 3). Both diverse public health recommendations and diet scores do not have the same restrictions concerning grains and dairy products [35,53,54,55,56]. Moreover, consumption of whole grains and low-fat dairy products, and thus detail of FFQs, may depend on both the cultural [57,58] and generational [57,59,60] habits. Lastly, a statistical requirement was needed to discriminate consumption. Indeed, had we adhered to Fung’s classification of food components proposed previously [10], the low number of food items included in some food components, the number of response categories, and the high number of non-consumers would have resulted in non-discriminating consumption frequencies for several cohorts.

### 4.2. Daily Frequency Consumption

Besides some differences between cohorts, consumption of each food component seems to be comparable with those reported in previous literature. The European Prospective Investigation into Cancer and Nutrition (EPIC) reported overall similar consumption of most food groups in France, Netherlands, and United Kingdom as we observed in the ALPHABET consortium cohort studies. For example, in EPIC the recorded mean consumption of vegetables, fruits, and red and processed meats was 128–261, 170–274, and 50–78 g/day respectively [61,62], we observe median consumption of 147–332, 285–360, and 51–87 g/day for the same food groups in the ALPHABET consortium cohorts. Within the Australian Longitudinal Study on Women’s Health, a country with a Western culture and presumably similar dietary habits, median intake values of breads/cereals, fruits, vegetables and dairy were 2.6, 2.2, 2.1, and 2.0 servings/day, respectively [63]. These values are comparable with those in the ALPHABET consortium cohorts: median intakes range from 1.6 to 3.5 servings/day for total grains, from 1.1 to 3.1 servings/day for fruits, from 1.5 to 3.0 servings/day for vegetables (excluding potatoes and condiments), and from 0.8 to 2.2 servings/day (except Lifeways with 0.1 servings/day) for non-full-fat dairy products.

A study comparing four established DASH diet indices (in association with colorectal cancer) suggested that all indices capture an underlying construct inherent in the DASH dietary pattern [38]. A consequence of the quintile-based design used in Fung’s scoring system and our DASH score, (determined by the specific cohort distribution of each food component), is that it discriminates individuals within each cohort but not necessarily between cohorts. Hence, identical scores in different cohorts may reflect similar dietary quality but could also arise from differences in FFQ characteristics (diverse lengths and details) across cohorts. Moreover, methodological choices such as caloric exclusion (e.g., Generation R) might generate cohort variability. However, the main aim in this study is to create a DASH score for all cohorts within our consortium using harmonised dietary data to reduce heterogeneity. To this end, the quintile approach is less prone to measurement error (as FFQs were used in all our cohorts) and the resulting cohort-specific DASH score should be a valid tool for intra-cohort ranking of dietary quality, which can be used for cohort-specific association analysis with health outcomes and subsequent meta-analysis.

### 4.3. Dietary Data Utilisation

Each cohort had its own protocol for measuring food intake. Indeed, the FFQs varied in length (from 43 to 293 food items), level of detail (from 5 to 9 response categories), and portion sizes used in the semi-quantitative FFQ (country- or region-specific). Such differences likely result in discrepancies in estimated intake. However, DASH scores incorporate a significant part of each FFQ: 48.1 to 79.1% of food items included in the FFQs (without alcohol); and the majority of food components comprise a significant number of food items. Indeed, except for a limited number of groups in ALSPAC and REPRO_PL, all food components relied on at least 5 food items. In addition, correlations between DASH scores calculated from frequencies and amounts (for cohorts with data available in both units) were very high. Thus, the method to derive the DASH score is robust and insensitive to country specific cultural habits which could impact portion sizes. Although FFQs have limitations such as recall or reporting biases, most of the FFQs were validated within each cohort in ALPHABET, and are, as already demonstrated in former studies, an appropriate tool to rank women based on energy, nutrients, and food intakes [64,65,66].

## 5. Conclusions

In conclusion, we developed a DASH score composed of eight food components and adapted to the ALPHABET consortium’s specificities. This work was essential for conducting future meta-analyses within the ALPHABET consortium and will permit us to explore the interplay between maternal diet quality, epigenetics, and offspring health. We demonstrated the feasibility of harmonising existing dietary intake data from diverse studies. The explanations of the method used to derive DASH scores and the challenges faced during this process may be useful to guide other researchers in adapting the DASH score to their study’s specificities.

## Figures and Tables

**Figure 1 nutrients-11-02706-f001:**
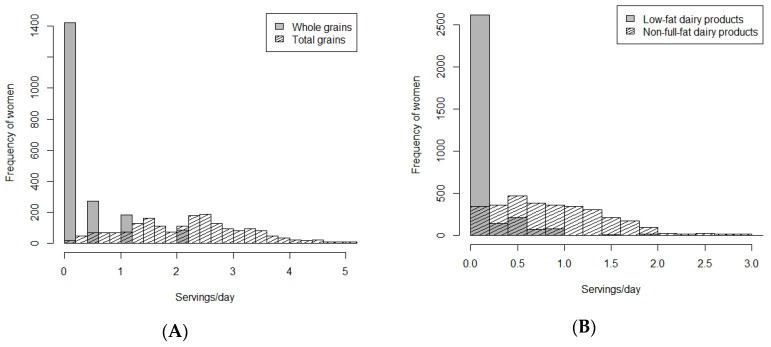
Distributions of servings/day (at pre-pregnancy period) between (**A**) whole grains and total grains in EDEN; and (**B**) low-fat and non-full-fat dairy products in SWS.

**Figure 2 nutrients-11-02706-f002:**
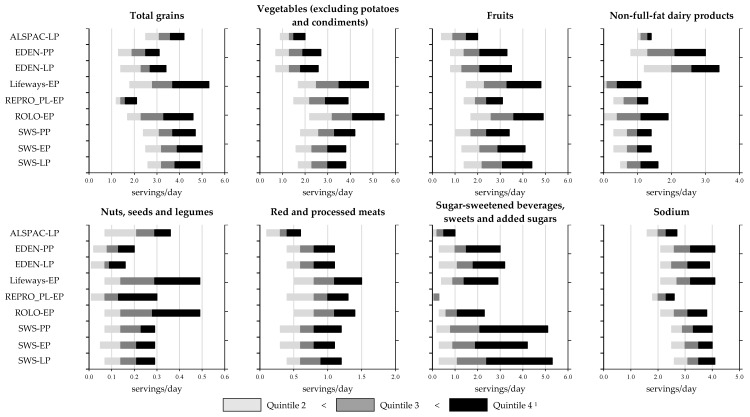
Maternal daily frequency consumption (in quintile) of the eight DASH food components by time point and cohort^1^ in ALPHABET. PP: pre-pregnancy, EP: early pregnancy, LP: late pregnancy. Generation R data are only available in amount and thus not presented. ^1^ The quintiles 1 and 5 spread from the minimum to the quintile 2 limit and from the quintile 4 limit to the maximum, respectively. Maximum observed is likely to exceed x-abscissa and is not necessary the maximum presented in abscissa axis.

**Table 1 nutrients-11-02706-t001:** Characteristics of the cohorts in the ALPHABET consortium.

Cohort	ALSPAC	EDEN		Generation R	Lifeways	REPRO_PL	ROLO	SWS		
**Number of Recruited Women**	14,541	2002		9778	1132	1451	759	12,583		
Women age eligibility	No age limit	18 and over		No age limit	No age limit	No age limit	18 and over	20–34 years		
Study type	Mother-child cohort	Mother-child cohort		Pregnancy-child cohort	Mother-child cohort	Mother-child cohort	Randomised control trial	Pre-pregnancy and pregnancy-child cohort		
Period of inclusion	1990–1992	2003–2006		2002–2006	2001–2003	2007–2011	2007–2011	1998–2002		
Location (specific cities)	England (Bristol)	France (Multicentre)		The Netherlands (Rotterdam)	Republic of Ireland (Multicentre)	Poland (Multicentre)	Republic of Ireland (Dublin)	England (Southampton)		
Assessment types	Non-quantitative FFQ	Semi-quantitative FFQ		Semi-quantitative FFQ	Semi-quantitative FFQ	Non-quantitative FFQ	Semi-quantitative FFQ	Non-quantitative FFQ		
Period of FFQ assessment	Around 32 WG	24–28 WG	Birth	<24 WG	12–16 WG	20–24 WG	≤28 WG	PP	11 WG	34 WG
FFQ window period	LP	PP	LP	EP	EP	EP	EP	PP	EP	LP
Number of women with validated FFQ ^1^	11,965	1964	1849	6402 ^2^	1121	1314	631	3156 ^3^	2270	2649
Mode of FFQ assessment	Self-reported	Self-reported		Self-reported	Self-reported	Self-reported	Self-reported	Nurse administered		

FFQ: Food Frequency Questionnaire. WG: weeks of gestation. PP: pre-pregnancy, EP: early pregnancy, LP: late pregnancy. ^1^ Finally included in this study. ^2^ Generation R used a caloric cut-off to exclude women with caloric intakes <500 kcal or >3500 kcal (all other cohorts did not make any exclusion based on energy before deriving the DASH score). ^3^ 12,572 women answered the FFQ at PP period but only 3158 women went on to have live singleton birth within the study.

**Table 2 nutrients-11-02706-t002:** Availability of FFQ data for each cohort.

Cohort	ALSPAC	EDEN	Generation R	Lifeways	REPRO_PL	ROLO	SWS	ALPHABET ^1^
FFQ total of food items	43	137	293	158	66	158	104	137.0
FFQ total of food items without alcohol	43	130	283	154	62	154	99	132.1
Total of food items selected for the DASH	34	65	136	85	36	85	58	71.3
% ^2^ items selected/total food items without alcohol	79.1%	50.0%	48.1%	55.2%	58.1%	55.2%	58.6%	57.8%
**Food components with higher intakes recommended**								
Whole grains	3	1	13	7	2	7	4	5.3
Total grains ^3^	7	7	20	14	5	14	8	10.7
Vegetables	5	16	33	24	12	24	16	18.6
Fruits	3	12	20	13	10	13	12	11.9
Low-fat dairy products	2	4	10 ^4^	6	0	6	2	4.3
Non-full-fat dairy products ^3^	3	6	18 ^4^	7	2	7	5	6.9
Lean meats, poultry, fish	4	9	10	9	13	9	5	8.4
Nuts, seeds, legumes	7	4	14	5	2	5	2	5.6
**Food components with lower intakes recommended**								
Fats and oils	5	10	13	15	2	15	11	10.1
Sweets and added sugars	3	5	9	4	1	4	3	4.1
Sugar-sweetened beverages	2	3	2	1	0	1	2	1.6
Red and Processed meat	4	12	20	17	4	17	10	12.0
Sodium	Available in grams/day	Available in grams/day	Available in grams/day	Available in grams/day	Available in grams/day	Available in grams/day	Available in grams/day	Available in grams/day

FFQ: Food Frequency Questionnaire. ^1^ Mean values (rounded to one decimal point) in the ALPHABET consortium. ^2^ Percentage (rounded to one decimal point). ^3^ With “whole grains” and “fat-free and low-fat dairy products” included respectively. ^4^ By combining items on foods and items on types of milk consumed.

**Table 3 nutrients-11-02706-t003:** Final food component choices.

Fung’s DASH Components	Original DASH Components	Food Components Selected in ALPHABET	Rationale
Whole grains	Total grains (Additional note: *“whole grains are recommended for most grain servings as a good source of fiber and nutrients”*)	Total grains	Consumption of grains is not explicitly limited by some public health organizations (e.g., PNNS ^1^ in France [43,44]) and some diet score are considering total grains (e.g., several Mediterranean diet [45]). The original DASH considers grains as a food component (foster whole grains are recommended as additional note only) [41]. Lack of information on whole grains in several ALPHABET cohorts (depending on FFQ length and detail) (e.g., EDEN has 7 items for total grains but a single item for whole grains).
Vegetables without potatoes	Vegetables	Vegetables (excluding potatoes and condiments)	Potatoes are not considered as vegetables according to Eurocode 2 [42]. By using servings/day, including condiments would generate an overestimation of the consumption.
Fruits	Fruits	Fruits	Not applicable
Low-fat dairy products	Low-fat milks & milk products	Non-full-fat dairy products	Some diet scores do consider total dairy products [40]. Equivocal scientific justification: *“the recommendation to focus on low-fat in place of regular- and high-fat dairy is currently not evidence-based”* [46]. Lack of information on low-fat dairy products in several ALPHABET cohorts (e.g., REPRO_PL does not have variables for low-fat dairy products).
Nuts, seeds, legumes	Nuts, seeds, legumes	Nuts, seeds, legumes	Not applicable
Red and processed meats	Lean meat, poultry, fish	Red and processed meats	Not applicable
	Fats and oils		
Sugar-sweetened beverages	Sweets and added sugars	Sugar-sweetened beverages, sweets, and added sugars	Take into Nuts, seeds, legumes account a wider diversity of sugar sources.
Sodium	Additional note for reducing salt	Sodium	Not applicable

FFQ: Food Frequency Questionnaire. ^1^ PNNS: Programme National Nutrition Santé (French National Nutrition and Health Program).

**Table 4 nutrients-11-02706-t004:** Median intakes of DASH food components by cohort and period in the ALPHABET consortium.

	ALSPAC		EDEN				Generation R		Lifeways		REPRO_PL		ROLO		SWS					
	LP		PP		LP		EP		EP		EP		EP		PP		EP		LP	
	Median[IQR]		Median[IQR]		Median[IQR]		Median[IQR]		Median[IQR]		Median[IQR]		Median[IQR]		Median[IQR]		Median[IQR]		Median[IQR]	
**Total grains**																				
f/d ^1^	3.4	[1.4]	2.3	[1.5]	2.5	[1.7]	-	-	3.2	[2.7]	1.6	[0.7]	2.8	[2.3]	3.4	[1.9]	3.5	[2.0]	3.5	[1.7]
g/d ^2^	-	-	142	[89]	149	[91]	162	[104]	162	[138]	-	-	183	[136]	-	-	-	-	-	-
**Vegetables (excluding potatoes and condiments)**																				
f/d	1.5	[0.9]	1.6	[1.6]	1.5	[1.4]	-	-	3.0	[2.5]	2.5	[1.8]	3.6	[2.4]	2.9	[1.9]	2.6	[1.8]	2.6	[1.7]
g/d	-	-	176	[201]	154	[167]	147	[89]	269	[222]	-	-	332	[224]	-	-	-	-	-	-
**Fruits**																				
f/d	1.1	[1.1]	1.7	[2.0]	1.7	[2.1]	-	-	2.8	[2.7]	2.2	[1.4]	3.1	[2.5]	2.1	[2.0]	2.5	[2.2]	2.7	[2.5]
g/d	-	-	311	[353]	293	[385]	285	[264]	340	[327]	-	-	360	[291]	-	-	-	-	-	-
**Non-full-fat dairy products**																				
f/d	1.3	[0.3]	1.6	[1.5]	2.2	[1.8]	-	-	0.1	[1.0]	1.0	[0.7]	1.0	[1.4]	0.8	[0.7]	0.8	[0.8]	1.0	[1.0]
g/d	-	-	303	[357]	417	[403]	224	[308]	18	[198]	-	-	250	[298]	-	-	-	-	-	-
**Nuts, seeds, legumes**																				
f/d	0.3	[0.3]	0.1	[0.1]	0.1	[0.1]	-	-	0.2	[0.3]	0.1	[0.3]	0.2	[0.4]	0.2	[0.2]	0.2	[0.2]	0.2	[0.2]
g/d	-	-	13	[16]	10	[17]	13	[19]	19	[49]	-	-	19	[21]	-	-	-	-	-	-
**Red and processed meats**																				
f/d	0.4	[0.3]	0.7	[0.6]	0.7	[0.5]	-	-	0.9	[0.8]	1.0	[0.7]	0.9	[0.8]	0.7	[0.7]	0.7	[0.6]	0.7	[0.7]
g/d	-	-	61	[57]	60	[55]	51	[44]	87	[66]	-	-	87	[67]	-	-	-	-	-	-
**Sugar-sweetened beverages, sweets, and added sugars**																				
f/d	0.4	[0.8]	1.2	[1.9]	1.3	[2.3]	-	-	1.1	[1.9]	0.3	[0.2]	0.9	[1.4]	1.2	[3.8]	1.2	[3.1]	1.7	[3.8]
g/d	-	-	49	[153]	55	[187]	67	[110]	46	[101]	-	-	28	[66]	-	-	-	-	-	-
**Sodium**																				
mg/d	2161	[854]	2871	[1636]	2831	[1461]	3324	[1298]	3001	[1512]	2163	[704]	2883	[1381]	3142	[1188]	3246	[1211]	3294	[1157]

PP: pre-pregnancy, EP: early pregnancy, LP: late pregnancy. IQR: Interquartile range. -: not available. ^1^ f/d: daily frequency in servings per day (rounded to one decimal place). ^2^ g/d: daily amount in grams and/or millilitres per day (rounded to the nearest whole number). mg/d: daily amount in milligrams per day (rounded to the nearest whole number).

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
