# Peer review of "Deriving the Dietary Approaches to Stop Hypertension (DASH) Score in Women from Seven Pregnancy Cohorts from the European ALPHABET Consortium"

_nutrients, 2019, doi:10.3390/nu11112706_

Round 1

Reviewer 1 Report

Dear Author(s)

This study will contribute to future studies exploring long-term interaction between maternal diet, epigenetics and neonatal/children’s health. It will further provide insight into the use of DASH in large-scale multicenter/multisite studies as well as avoiding challenges in harmonizing data related to DASH. The introduction provides a brief understandable content by outlining the context on maternal diet & its connection with later Maternal/neonatal/child health, approaches to the use of DASH and overview of the ALPHABET consortium. The study aim/objective was clear on what the study aims to achieve. Similarly, the conclusion is justified with respect to the results presented. My few concerns are as follows;

While it is commendable that the study tries to put together a large volume of data, it may be difficult for a non-expert reader to conceptualize the outcome of the study from the abstract. It is worthy to make the abstract as clear/concise as possible. The abstract set out the study context and how the first objective (to explain the method used to derive a harmonised DASH score in the ALPHABET consortium and detail the challenges encountered during this process) was achieved. However, a summary of outcome with respect to the second objective (to describe the maternal DASH score and related food consumption in different cohorts and over three periods of assessment) wasn’t clearly outlined in the abstract. It could be improved by adding a brief finding with respect to maternal dash scores or the highest and/or lowest food-related consumption score at pre-pregnancy, early pregnancy and late pregnancy across the cohorts. I wonder if it’s possible to harmonise the categories of ‘pre-pregnancy, early pregnancy (1st or 2nd trimester) and late pregnancy (3rd trimester)’ with that of ‘PP: pre-pregnancy, EP: early pregnancy, MP: middle pregnancy, LP: late pregnancy’ in table 1 by adopting one category that is more appropriate throughout the manuscript. That would aid understanding and flow of information.

Good manuscript.

Reviewer 2 Report

In their research output, Aubert  et co-workers derived DASH scores for pregnant women from seven European birth cohorts after harmonising data from FFQs of various lengths and degrees of detail. The reason was related to the need to examine the interplays between maternal diet quality, epigenetics and offspring health by using a similar score among settings. They detailed the methodology, the difficulties and the solutions in the management of the derivation of the DASH score. The topic is not original, but treated over the years, without reaching any strong consensus. This research has the merit to harmonize the previous published material on this matter. It is well written and clear in their sections. Its readibility might seem difficult in some technical points for a medium reader due to the detailed methodology applied. The conclusions are consistent with the evidence and data presented, reaching the main objective. The efforts seem to be of interest, although with the limitations acknowledged by the same Authors. Sometimes we need to go out the limitations, if we want to improve our knowledge.
